# Cofilin and Neurodegeneration: New Functions for an Old but Gold Protein

**DOI:** 10.3390/brainsci11070954

**Published:** 2021-07-20

**Authors:** Tamara Lapeña-Luzón, Laura R. Rodríguez, Vicent Beltran-Beltran, Noelia Benetó, Federico V. Pallardó, Pilar Gonzalez-Cabo

**Affiliations:** 1Department of Physiology, Faculty of Medicine and Dentistry, University of Valencia, 46010 Valencia, Spain; luzon.lapena@uv.es (T.L.-L.); Laura.Robles@uv.es (L.R.R.); vibelbel@alumni.uv.es (V.B.-B.); noeliabg92@gmail.com (N.B.); federico.v.pallardo@uv.es (F.V.P.); 2Biomedical Research Institute INCLIVA, 46010 Valencia, Spain; 3CIBER de Enfermedades Raras (CIBERER), 46010 Valencia, Spain

**Keywords:** cofilin, neurodegenerative diseases, cofilin–actin rods, apoptosis, mitochondrial fission, microtubule instability

## Abstract

Cofilin is an actin-binding protein that plays a major role in the regulation of actin dynamics, an essential cellular process. This protein has emerged as a crucial molecule for functions of the nervous system including motility and guidance of the neuronal growth cone, dendritic spine organization, axonal branching, and synaptic signalling. Recently, other important functions in cell biology such as apoptosis or the control of mitochondrial function have been attributed to cofilin. Moreover, novel mechanisms of cofilin function regulation have also been described. The activity of cofilin is controlled by complex regulatory mechanisms, with phosphorylation being the most important, since the addition of a phosphate group to cofilin renders it inactive. Due to its participation in a wide variety of key processes in the cell, cofilin has been related to a great variety of pathologies, among which neurodegenerative diseases have attracted great interest. In this review, we summarized the functions of cofilin and its regulation, emphasizing how defects in these processes have been related to different neurodegenerative diseases.

## 1. Introduction

In 1980, Bamburg et al. identified a protein that acted as an actin disassembly factor in chicken brain extracts [1]. Proteins with similar functions were isolated from different organisms and tissues during the following years [2] and were grouped under the ADF/cofilin family, which includes actin-depolymerizing factor (ADF, also known as destrin); cofilin-1, the major ubiquitous isoform in non-muscle tissues; and cofilin-2, the major isoform in differentiated muscle. In this review, we will focus on cofilin-1 (hereinafter called cofilin).

Actin microfilaments (F-actin) are linear polymers composed of globular actin monomers (G-actin), which are polarized with a barbed end, where the addition of available actin monomers bound to ATP occurs, and a pointed end, where actin bound to ADP is released [3]. The addition and release of actin monomers allows actin microfilaments to be a dynamic structure capable of responding to stimuli. These actin dynamics are possible thanks to accessory proteins such as the Arp2/3 complex and formin, involved in actin nucleation, a process of the formation of a complex composed of actin monomers from which an actin filament can elongate; profilin, involved in microfilament elongation; or ADF/cofilin, involved in actin disassembly, although its function has been shown to vary depending on cofilin concentration relative to actin [4], as Figure 1 shows. At low cofilin/actin ratios, cofilin binds to the ADP-actin region of the F-actin, and it severs filaments in a persistent way creating new barbed and pointed ends. Then, cofilin is released from binding to an ADP-actin subunit and both can be recycled. At the same time, the pieces of F-actin generated by severing can nucleate filament growth or enhance depolymerization if ATP-actin is limited [5]. At higher cofilin/actin ratios, many cofilin subunits bind to ADP-actin, induce conformational changes in the microfilament modifying its twist and sever it in a rapid, but not persistent, way as most cofilin is sequestered and bound to actin. Cofilin saturates the fragments, which are stabilized, although they can be depolymerized to generate monomers or may be used to nucleate growth [4]. Under ATP-deficient conditions, stabilized actin bound to cofilin can form rods, slowing the actin dynamics and associated ATP hydrolysis. This mechanism allows the preservation of ATP, but if it persists, the rods can cause neurite degeneration (reviewed below) [6]. When even higher cofilin/actin ratios occur, cofilin can bind actin monomers and nucleate new filaments promoting polymerization [7].

However, cofilin has been recently related to cellular processes other than actin microfilament disassembly. For example, cofilin could take part in rod formation, apoptosis induction, mitochondrial dynamics, microtubule instability, or even in the regulation of the gene expression [8,9,10]. Alterations in these processes could trigger pathological conditions such as neurodegenerative diseases. Neurons are cells susceptible to cytoskeleton disturbances that can lead to deficiencies in axonal transport. In addition, neurons present a high energetic demand, which, to be satisfied, requires that mitochondrial dynamics works properly [11,12,13]. However, not only neurons are involved in the correct functioning of the nervous system. The role of glial cells is essential by providing support to neurons in relevant processes including synapses, neuronal plasticity, brain fluid transport mediated by the glymphatic system, and the inflammatory response [14,15,16,17]. As in neurons, cofilin could cause alterations in glial functions or morphology changes and, consequently, lead to neurodegenerative diseases [18]. In this regard, great advances have been made to discover how cofilin and its regulatory mechanisms may be involved in neurodegenerative diseases such as Alzheimer’s disease (AD), Huntington’s disease (HD), or Parkinson’s disease (PD) [19,20,21]. In this review, we analyse the main mechanisms through which cofilin, and its regulation could be taking part in the pathology of different neurodegenerative diseases (Figure 2).

## 2. Cofilin Regulation and Its Implication in Neurodegenerative Diseases

Cofilin can undergo post-translational modifications that regulate its function and, among these, phosphorylation stands out. Phospho-regulation of cofilin is mediated by signalling pathways in response to extracellular signals. This signal transduction is very complex, and numerous studies have been carried out to characterize it [22]. Cofilin is inactive when it is phosphorylated on its Ser3 residue by LIM-domain containing kinases 1 and 2 (LIMK1 y LIMK2) and testis-specific kinases 1 and 2 (TESK1 y TESK2). LIMK1 is in turn activated by the small GTPase Rho and its downstream Rho-associated protein kinase (ROCK) [23], and TESK1 is activated downstream of an integrin signal. Both contribute to stabilize the actin cytoskeleton. The phosphatases chronophin (CIN) and slingshot (SSH1) remove the phosphate of Ser3, activating cofilin and accelerating the dynamics of the actin cytoskeleton [24]. However, in addition, the regulation of cofilin has to be spatiotemporally controlled to obtain a precise actin cytoskeleton reorganization. While cofilin is expressed in almost all tissues and cells, distinctly different expression patterns of LIMKs, TESKs, and SSHs exist [22].

Moreover, other regulatory mechanisms of cofilin are known. Dephosphorylation by CIN and SSH is enhanced when ATP levels are low and reactive oxygen species (ROS) high [25,26], demonstrating how oxidative stress plays a role in the regulation of cofilin activity. Moreover, cofilin function is also directly regulated by the oxidation of its cysteine residues [27]. Oxidized cofilin can induce the formation of an intramolecular disulfide bridge in vivo and the loss of its Ser3 phosphorylation site. This activated and oxidized cofilin is responsible for inducing the apoptosis cascade through its translocation to the mitochondria [28]. In addition, cofilin oxidation can induce not only intramolecular but also intermolecular disulfide bonds between cofilin molecules. In these cases, cofilin–actin rod formation is enhanced, thus preventing the cell from entering apoptosis [29]. Apart from phosphorylation and oxidation, other regulatory mechanisms have been proposed. For example, phosphatidylinositol 4,5-bisphosphate (PIP2) is a membrane phospholipid that inhibits cofilin activity by competing with actin at the binding site [30]. High-pH conditions could also regulate the activity of cofilin by increasing it [31]. Cofilin phosphorylation at its Y68 residue induces cofilin ubiquitination and degradation through the proteasome, adding another regulation mechanism [32].

According to current evidence, it seems clear that cofilin is involved in neurodegenerative processes, as we will describe in the following paragraphs, and more specifically, the regulation of cofilin function has also been associated with different disease states. However, much remains to be known about the complex regulatory mechanisms of this protein and how they are affected in different neurodegenerative diseases.

Mutations in the Parkin 2 (*PARK*) gene are the main cause of autosomal recessive parkinsonism, a type of early onset familial PD, and also seem to play a role in sporadic PD. Parkin is a ubiquitin E3 ligase that ubiquitinates dysfunctional proteins for their degradation in the proteasome. The loss of parkin activity leads to the accumulation of parkin substrates. These have been proposed as a cause for toxicity and neurodegeneration through mechanisms that are not completely understood [33]. Previous studies had already shown that parkin was associated with actin filaments by colocalizing with them, so it was suggested that parkin could have a role in actin stabilization [34]. According to this finding, Lim et al. observed that parkin interacts with and ubiquitinates LIMK1 in the human dopaminergic neuroblastoma cell line BE(2)-M17 but not in the human embryonic kidney-derived cell line HEK293, indicating a tissue-specific regulation. As a result, since LIMK1 phosphorylates cofilin, LIMK1 ubiquitination by PARK2 reduces the level of inactive cofilin [35]. Overall, parkin modulates the level of phosphocofilin by negatively regulating LIMK1 activity in a cell-type-dependent way, which can help understand how cofilin contributes to familial PD.

Friedreich’s ataxia (FRDA) is an inherited peripheral neuropathy characterized by an early loss of neurons in the dorsal root ganglia, among other clinical symptoms caused by frataxin deficiency. Cytoskeletal abnormalities have been proposed to contribute to dying-back neurodegeneration in FRDA [24]. In fact, at the molecular level, the F-actin:G-actin ratio has been shown to be increased in sensory neurons of FRDA mice, suggesting an alteration of the normal turnover of actin filaments. This disturbance could be the cause of the changes observed in the morphology of the growth cones of FRDA mouse neurons [36]. The authors observed a hyperactivation of cofilin in the dorsal nerve roots of the FRDA mouse model compared to controls, which can be partially explained by the increased levels of CIN [36]. Altogether, the dysregulation of cofilin might explain the reduced neurite growth and alterations of cytoskeleton, suggesting a role in FRDA neuropathy.

Regulation of active cofilin could also be involved in AD. A significant reduction of total SSH1 phosphatase in AD brains could be the cause of the increased cofilin inactivation observed in human samples. γ-Secretase is an enzyme that makes the second cleavage to amyloid precursor protein (APP) to give rise to the amyloid-β (Aβ) peptides, whose pathological accumulation is related to AD. This enzyme has been discovered to takes part in promoting actin/cofilin-related pathology. In this sense, γ-secretase inhibitors promote cofilin activation by increasing SSH1 levels in mouse primary cortical neurons [37]. However, other authors point to the excessive cofilin activation, and not the opposite, as the cause of AD pathogenesis. For example, Kim et al. suggest that the binding of Aβ oligomers to the leukocyte immunoglobulin-like receptor B2 (LilrB2) present in human brain enhances cofilin signalling, which leads to synapse elimination [38].

The contribution of active cofilin to AD pathogenesis does not seem to be limited only to its levels since its location also has an influence. Cofilin is translocated to the spine upon long-term potentiation induction and promotes the assembly of F-actin, essential for spine expansion and to potentiate synaptic transmission. This activity-dependent plasticity phenomenon is altered in the hippocampus of AD patients, with cofilin having a potential role in this defect since it is aberrantly localized in neuron spines [39]. The authors showed how downregulation of cyclase-associated protein 2 (CAP2), by synergizing with cofilin to accelerate depolymerization of the pointed end of actin filaments, is able to control cofilin synaptic availability in long-term potentiation processes that, as it is known, are extremely important for stabilizing memory.

It seems clear that the regulation of cofilin is crucial for the activity and function of the protein, so that alterations in the cofilin regulation mechanisms trigger pathological processes such as neurodegenerative diseases.

## 3. New Functions for Cofilin in Neurodegenerative Diseases

Chua et al. have shown that after staurosporine-induced apoptosis, active cofilin can translocate from the cytoplasm to the mitochondria, leading to the opening of the mitochondrial permeability transition pore and the release of cytochrome C, which is the first step in the apoptotic cell death cascade [28]. This suggests that cofilin has an important function during the initiation phase of apoptosis. In this regard, the translocation of cofilin to mitochondria has been also linked to mitochondrial fission, a process in which cofilin would participate regulating the function of the dynamin-related protein 1 (Drp1) [40]. Moreover, cofilin could also be translocated to the nucleus due to its nuclear localization sequence [41]. In this context, cofilin has been suggested to participate in actin import to the nucleus, where actin can regulate gene expression [42]. Even inactive cofilin, a form that was thought to have no function, has been shown to have stimulatory effects on phospholipase D1 (PLD1), an enzyme involved in a wide variety of cellular responses, including calcium mobilization, superoxide production, endocytosis, vesicle trafficking, etc. [43].

All these processes, highly related to neurodegeneration, point towards cofilin as a key protein involved in many neurological diseases, but its role in the pathophysiology of these diseases still remains to be elucidated.

### 3.1. Cofilin Oxidation Leads to Cofilin–Actin Rod Formation

Cofilin–actin rods are inclusions that contain cofilin and actin in a 1:1 ratio, formed in axons and dendrites of stressed neurons. Rods can be assembled in response to different stimuli, including sublethal doses of glutamate, hydrogen peroxide, or ATP depletion, although all the mechanisms of rod formation converge in an increase in the pool of dephosphorylated/activated cofilin by the action of phosphatases [44]. In this context, cofilin binds with high affinity to ADP-actin, saturating it and forming rods, as Figure 1 shows. Actin in rods is less dynamic, which avoids the ATP hydrolysis associated with actin filament turnover. As a result, the formation of rods transiently protects neurons in declining ATP situations. Cofilin–actin rods can be reversible when the insult is ended but, if it persists, rods may block intracellular trafficking and induce synaptic disfunction, contributing to neurodegeneration and ageing [6,45]. In fact, rods were found in the hippocampal and cortical neurons of the post-mortem brains of patients with Alzheimer’s disease [44]. Maloney et al. proposed a feedforward model to explain neurodegeneration in AD in relation to cofilin–actin rods. The authors observed that both the pathological accumulation of Aβ peptides, which occurs in familial AD, and the neuronal stress (ischaemia, oxidative stress, excitotoxic insults, etc.) involved in sporadic AD can result in rod formation in the presence of phosphatases responsible for activating cofilin [46]. Rod formation can, in turn, block the transport of vesicles containing APP, β-secretase, and γ-secretase. These stalled vesicles supply a site for Aβ production, which triggers rod formation in surrounding neurons, exacerbating neurodegeneration [47].

Cofilin–actin rods have also been related to other neurodegenerative diseases such as HD. Some studies proposed that cofilin–actin rods can also be located inside the nucleus instead of in the cytoplasm under various stressful cellular conditions [48,49]. In fact, Munsie et al. described that, under stress conditions such as heat shock or DMSO-induced stress, huntingtin, the protein whose mutated form causes HD, was localized in the nuclear cofilin–actin rods in a mouse-neuron-derived cell line and in primary mouse hippocampal neurons. Moreover, once cells had recovered from stress, nuclear rods were persistent only when mutant huntingtin was present, suggesting that huntingtin could have a role in actin remodelling in response to stress [21].

### 3.2. Cofilin Takes Part in Microglia Activation Which Leads to Neuroinflammation

Neuroinflammation involves the activation of glial cells, especially microglia and astrocytes, which precedes and causes neuronal degeneration. In response to an increase in inflammatory cytokines, microglia and astrocytes undergo morphological and molecular changes that result in glial reactivity.

Cofilin participates in the morphological changes of microglia and astrocytes in response to neuroinflammatory process [18]. Lipopolysaccharide (LPS)-induced microglial cell activation was studied in a cofilin-knockdown cellular model, confirming a reduction in microglial activation and the protection of neurons from neurotoxicity [50]. The authors propose that LPS, by binding to the toll-like receptor 4 (TLR4), activates the SRC kinase that most likely promotes cofilin dephosphorylation through the SHH phosphatase. Thus, morphological changes, proliferation, migration, and the phagocytosis of microglial cells were altered. Moreover, cofilin could also act on the transcription factors STAT1 and NF-kB, which, in turn, initiate neurotoxicity through the expression of cytotoxic factors [50].

After a large inflammatory response, microglia adopt a phagocytic phenotype (neuroprotective phenotype) for the removal of cellular debris. It is in this action where cofilin intervenes, promoting the cytoskeletal dynamics and leading to a change to ameboid morphology and phagocytic activity of the microglia [51].

### 3.3. Cofilin Mediates Actin Depolymerization for Myelin Wrapping

The formation of myelin is important for the correct function of motor and sensory neurons. During the development of the central nervous system, oligodendrocyte precursor cells are responsible for extending a network of processes to contact axons and initiate myelination [52]. In this action, the dynamics of the cytoskeleton is vital, and any alteration in both the formation and repair of myelin generates neurodegenerative diseases such as AD, multiple sclerosis, amyotrophic lateral sclerosis, or psychiatric diseases [53,54,55].

It is well described that the initiation of myelination requires actin polymerization, while depolymerization is associated with myelin wrapping [52]. In this process, the cofilin must move to the oligodendrocyte membrane mediated by myelin basic protein (MBP), which facilitates actin depolymerization and, therefore, the wrapping process [56].

A protein necessary for the myelination and myelin repair processes is cyclin-dependent kinase 5 (CDK5). The Cdk5 KO mouse exhibits hypomyelination and a reduction in the number of Ranvier’s nodules. Molecular studies showed that low levels of p-cofilin together with p-CREB were associated with these phenomena, suggesting that these pathways were affected by low levels of myelin [57]. Recently, it has been observed that the regulation of actin polymerization during the initiation of myelination is determined by mTOR [58]. In these studies, the mTOR inhibitor rapamycin was added to primary oligodendrocyte cultures, observing a decrease in inactive cofilin levels while total cofilin levels did not change. Nevertheless, in addition, signalling after mTOR inhibition reduces both MBP levels and its correct location in the membrane, which directly affects myelin wrapping [58].

### 3.4. Cofilin Translocation into the Mitochondria Induces Apoptosis

As aforementioned, upon oxidative stress conditions, active cofilin becomes oxidized on its cysteine residues and, consequently, cofilin is translocated into the mitochondria. In this organelle, oxidized cofilin induces apoptosis through the release of cytochrome C. HTT2 cells (hippocampus-derived neuroblastoma cell line) treated with Aβ oligomers showed an increased translocation of cofilin to the mitochondria. However, the translocation was significantly prevented with the RNAi-mediated knockdown of the phosphatase SSH1, suggesting that Aβ oligomers promote apoptosis trough the activation and translocation of cofilin into the mitochondria via SSH1. The same authors isolated the mitochondrial fraction from the frontal cortex of AD patients and compared it with age-matched cognitively normal controls. They observed a significant increase in total and oxidized cofilin oligomers, as well as in cofilin monomers, reinforcing the idea that activated and oxidized mitochondrial cofilin has a pathogenic role in AD [59]. The mechanism by which Aβ oligomers promote actin–cofilin pathology in AD by both promoting rod formation and apoptosis is not fully understood yet. RanBP9, a protein known to promote Aβ oligomers production by scaffolding LRP/APP/integrins complexes, could take part in this process because RanBP9 overexpression promotes Aβ-induced neurotoxicity and apoptosis, which is independent of its capacity to promote Aβ generation [60]. Moreover, RanBP9 promotes cofilin activation via activating phosphatase SSH1. In transgenic mice exhibiting reduced RanBP9 protein, significant decreases in SSH1 protein could be observed, indicating that RanBP9 positively regulates SSH1 levels [61]. Another study has observed that active and oxidized cofilin, through Aβ accumulation and oxidative stress, binds p53 protein and promotes the translocation of this complex into the mitochondria. Once this complex is translocated, it activates the mitochondria-mediated apoptosis pathway. Meanwhile, cofilin enhances F-actin depolymerization, which would lead to p53 translocation into the nucleus according to previous studies that showed that actin polymerization impairs p53 nuclear import. Nuclear location of p53 would drive the transcription of the pro-apoptotic mitochondrial protein, suggesting that cofilin is involved in both direct mitochondrial and indirect nuclear apoptosis. SSH1 activation by RanBP9 could be an upstream factor that is involved in the cofilin–p53 apoptosis pathway. Moreover, the same study shows that inactive cofilin can activate PLD1, which inhibits cofilin/p53 complex formation and p53 translocation to the nucleus. Thus, PLD1 negatively regulates the cofilin/p53-mediated apoptosis pathway [62].

### 3.5. Cofilin Induces Mitochondrial Fission

Mitochondrial dynamics (fission and fusion) is a key process controlling cell death, mitophagy, organelle distribution, transport, and bioenergetics in the cell [11]. Neurons require large amounts of energy to carry out their functions, so they are especially sensitive to changes in mitochondrial dynamics. In fact, failures in this process have been related to neurodegenerative diseases such as Charcot–Marie–Tooth, AD, PD, or HD [63].

Drp1 is a GTPase that constricts the mitochondrial outer membrane, enhancing mitochondrial fission. To carry out its function, Drp1 must be recruited at the mitochondrial surface in a process promoted by actin polymerization. Regarding this, in mouse embryonic fibroblasts, active cofilin would act as a negative regulator of mitochondrial Drp1 activity and, thus, of mitochondrial fission by fine-tuning actin dynamics at the mitochondrial surface [40]. However, other authors point to both active cofilin and Drp1 translocation to the mitochondria as necessary for mitochondrial fission. Hu et al. have shown that ROCK1 activation results in the dephosphorylation of Drp1 and cofilin via serine/threonine phosphatases (PP1/PP2) in cancer cells, leading to the mitochondrial translocation of Drp1 and cofilin, which results in mitochondrial fission [64]. Furthermore, mitochondrial fission would activate the apoptosis process due to the release of cytochrome C [65]. In PD, ROCK1 has been described to promote aberrant mitochondrial fission by inducing the dephosphorylation/activation of Drp1, resulting in dopaminergic nerve cell apoptosis [20]. Recently, an approach to treat neurodegeneration in PD by the conversion of astrocytes to new dopaminergic neurons, which provide axons to reconstruct the nigrostriatal circuit, has been developed [66].

### 3.6. Cofilin Mediates Microtubule Instability

Tauopathies are a type of neurodegenerative disorders characterized by the accumulation of abnormally hyperphosphorylated protein tau, a microtubule-associated protein, in neuronal/glial inclusions. Tau accumulation is found in Pick’s disease, frontotemporal dementia, or AD, in which it is a hallmark in addition to the presence of β-amyloid plaques [67]. Cofilin has been discovered to not only regulate the action of Aβ peptides in AD, as previously mentioned, but also to participate in the pathological accumulation of tau in the inclusions. In this sense, active cofilin competes with tau for direct microtubule binding, displacing it from microtubules and inhibiting tau-induced microtubule assembly. Displaced tau is then accumulated, and it can be phosphorylated by several kinases, promoting tauopathy [10,68].

### 3.7. Cofilin Regulates Gene Expression

For many years, the location of actin was open to debate since actin did not present a nuclear localization sequence and, therefore, presence in this organelle was assigned to cytosolic contamination caused due to its high expression levels. Many subsequent studies confirmed that the presence of actin in the nucleus was indeed real and that it even had a very relevant role in transcription and transcriptional regulation through a sophisticated mechanism in eukaryotic cells (for a review see [69]). Cells can reach a high level of gene expression regulation by tuning the state and levels of actin. It is at this point where cofilin comes into action by being responsible for the active transport of actin into the nucleus. Cofilin has a nuclear localization sequence, recognized by importin-9, which allows the translocation of both proteins from the cytosol to the nucleus [42]. Furthermore, some authors suggest that cofilin could actually have a direct role in the regulation of gene expression, since increasing nuclear actin levels independently of cofilin does not affect the transcription levels [42]. It remains unclear how actin and/or cofilin have an effect on transcriptional regulation, whether it has a global or a more specific effect. In this regard, some experiments have demonstrated that the effect could target specific gene programmes and lead to reprogramming and cellular differentiation [70]. It is interesting to emphasize that this same regulatory mechanism could be present at the mitochondrial DNA level [71], since the localization of both proteins in these organelles has been observed.

## 4. Future Perspectives

Cofilin has a main role in actin dynamics and, therefore, in cytoskeletal homeostasis. However, current evidence on its new functions seems to show that cofilin also contributes to degenerative processes through the formation of cofilin–actin rods that impair axonal transport and promotion of neuronal cell death, or by changes in mitochondrial dynamics and in the endoplasmic reticulum–mitochondria (ER–mitochondria) connection and communication. However, much remains to be known about cofilin, starting with its complex regulatory mechanisms. This is a key piece to understand how the modulation of cofilin activity and levels could be a critical point in neurodegenerative diseases to modify its natural history. At this point, the question arises as to whether cofilin could be used as a neurodegenerative progression biomarker. In this sense, cofilin 2 expression was demonstrated to be significantly increased in the serum of Alzheimer’s disease patients, and it performed well as a diagnostic and non-invasive biomarker with high sensitivity and specificity [72].

Thus, the biggest challenge is to determine whether cofilin is a potential target for neurodegenerative diseases, which can open new possibilities for drug development with the purpose, for example, of preventing mitochondrial alteration and axonopathy in neurodegenerative diseases. Another therapeutic approach could be to design drugs that target cofilin-regulatory proteins since many of the pathological processes related to cofilin are due to alterations in its regulation. Anyway, the recent advances in the use of high-throughput screening and computer-aided drug design can help in the search of treatments for neurodegenerative diseases [73].

## Figures and Tables

**Figure 1 brainsci-11-00954-f001:**
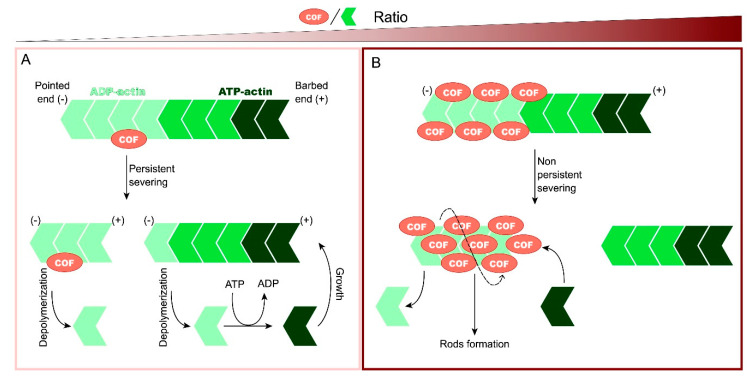
Function of cofilin in actin dynamics according to the cofilin/actin ratio. Cofilin has a pointed end, where actin subunits can be released from the microfilament, and a barbed end, where actin subunits can be added. (**A**) At low cofilin/actin ratios, cofilin binds to ADP-actin of filaments and severs it, creating new pointed and barbed ends. Then, cofilin binds to the generated fragments and induces depolymerization. Both cofilin and the released actin subunits can be recycled. (**B**) At higher cofilin/actin ratios, many cofilin subunits bind to ADP-actin, induce conformational changes in the microfilament, and sever it. However, severing is not persistent because cofilin is sequestered and bound to the microfilament. The generated F-actin fragments bound to cofilin are stabilized and can form rods under ATP-deficient conditions.

**Figure 2 brainsci-11-00954-f002:**
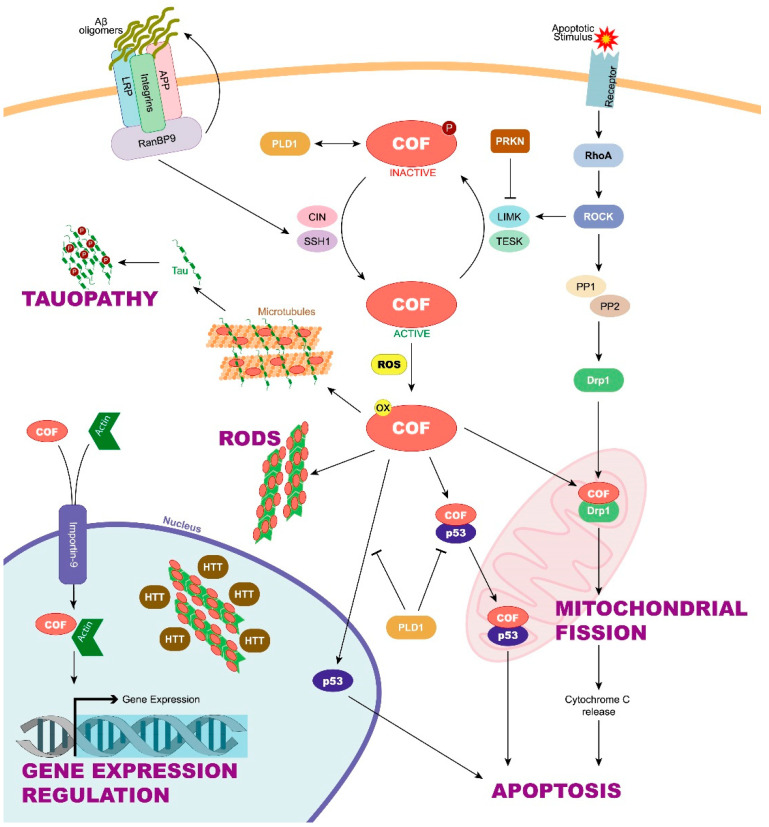
Role and regulation of cofilin in neurodegeneration. Cofilin function depends on the phosphorylation of its Ser3 residue, among others. When Ser3 is dephosphorylated by CIN and SSH1, cofilin enters an active state that can be enhanced by Aβ oligomer accumulation. On the other hand, LIMK and TESK can phosphorylate, and thus, inactivate cofilin. Parkin can interfere with cofilin phosphorylation by blocking LIMK function. Under oxidative stress conditions, active and oxidized cofilin can be translocated into the mitochondria, allowing it to take part in mitochondrial fission. This process is mediated by both active cofilin and the DRP1 protein and can be induced by activation of the ROCK pathway. Mitochondrial fission, in turn, triggers the release of the cytochrome C, leading to apoptosis. Apoptosis can also begin with the cofilin/p53 pathway, although inactive cofilin can prevent it activating PLD1. Moreover, active and oxidized cofilin can form cofilin–actin rods. The rods have been found to colocalize in the nucleus with huntingtin. Cofilin can also enter the nucleus together with actin through importin-9. Once in the nucleus, cofilin and actin can regulate gene expression. Finally, cofilin competes with tau for microtubule binding, displacing tau and promoting tauopathies. Aβ, amyloid β; APP, amyloid precursor protein; CIN, chronophin; COF, cofilin; Drp1, dynamin-related protein 1; HTT, huntingtin; LIMK, LIM-domain containing kinase; PLD1, phospholipase D1; PP1/PP2, protein phosphatase 1 and 2; PRKN, parkin; SSH1, slingshot; TESK, testis-specific kinase.

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
