# Peer review of "Cofilin and Neurodegeneration: New Functions for an Old but Gold Protein"

_brainsci, 2021, doi:10.3390/brainsci11070954_

Round 1

Reviewer 1 Report

The manuscript by Lapeña-Luzón et al discusses the role of cofilin and its regulation, emphasizing how defects in these processes have been related to different neurodegenerative diseases.

The review is well-summarised, informative and up-to-date (in most parts). Authors were successful in providing some well compiled opinions and summaries. The mechanistic figures can be a good starting point for future studies and will be of interest for Brain Sciences readers and beyond. 
However, there is a number of major and minor points that would need to be addressed in order to improve the quality of this paper before it can be accepted for publication.
Major:
- This review overlooked some essential and up-to-date work regarding the pathophysiology of neurodegenerative diseases (NDs), especially the role of glial cells and recent advances in their target validation and future therapies for NDs.
Authors should start by introducing the different brain cells, not only neurons as in lines 81-83. Also, they should add something about the energetic brain and how astrocytes play a role in what’s known as “Tripartite synapse”. References:
https://pubmed.ncbi.nlm.nih.gov/10322493/ 
https://pubmed.ncbi.nlm.nih.gov/31318452/ 

Neurodegenerative diseases are no longer neuron only disorders since the role of glial cells have been extensively validated. For example, the water channel aquaporin 4 (AQP4), which is highly expressed on astrocyte endfeet, critically regulates water flux between blood and brain. AQP4 plays an essential role in regulating the recently discovered glymphatic system which is a waste clearance system that utilizes a unique system of perivascular channels, formed by astroglial cells, to promote efficient elimination of soluble proteins and metabolites from the central nervous system. References:
https://www.ncbi.nlm.nih.gov/pmc/articles/PMC4252540/ 
https://pubmed.ncbi.nlm.nih.gov/30561329/ 
https://pubmed.ncbi.nlm.nih.gov/33004510/ 

Lines of evidence showed that CNS disorders, inflammation and vascular diseases caused by astrocytes, leading to brain edema and exacerbate neurodegenerative diseases. This has been recently been demonstrated by the breakthrough study by Kitchen et al Cell 2020 where they showed that targeting astrocytes following ischemia and hypoxia not only reduces edema but also stabilises the BBB/BSCB barriers and led to accelerated functional recovery compared with untreated animals. This role has been recently been confirmed by the work of Sylvain et al BBA 2021 which has demonstrated that targeting astrocytes is a viable therapeutic target using a photothrombotic stroke model. They have also shown a link to brain energy metabolism as indicated by the increase of glycogen levels. References:
https://pubmed.ncbi.nlm.nih.gov/32413299/ 
https://pubmed.ncbi.nlm.nih.gov/33561476/ 

Moreover, authors need to mention the work by Qian et al. Nature 2020 where they have beautifully shown that the conversion of midbrain astrocytes to dopaminergic neurons, which provide axons to reconstruct the nigrostriatal circuit. References:
https://pubmed.ncbi.nlm.nih.gov/32581380/ 

-End of discussion and towards the conclusion: NDs are yet incurable diseases. Author needs to point out to the recent advances in applying the use of high-throughput screening and computer-aided drug design as have been nicely reviewed by Aldewachi et al 2021 as they can provide a novel insight that can support target validation in future studies. References:
https://pubmed.ncbi.nlm.nih.gov/33672148/ 

Best.

Author Response

We are very grateful to the reviewer for their time and positive suggestions. The reviewer’s comments are shown in black and our replies are shown in red.

Reviewer 1:

The manuscript by Lapeña-Luzón et al discusses the role of cofilin and its regulation, emphasizing how defects in these processes have been related to different neurodegenerative diseases.

The review is well-summarised, informative and up-to-date (in most parts). Authors were successful in providing some well compiled opinions and summaries. The mechanistic figures can be a good starting point for future studies and will be of interest for Brain Sciences readers and beyond. 

We thank the reviewer for his/her kind comments and appraisal of our manuscript.

However, there is a number of major and minor points that would need to be addressed in order to improve the quality of this paper before it can be accepted for publication.

We have implemented all of them in the new draft of the manuscript.

Major:
- This review overlooked some essential and up-to-date work regarding the pathophysiology of neurodegenerative diseases (NDs), especially the role of glial cells and recent advances in their target validation and future therapies for NDs. Authors should start by introducing the different brain cells, not only neurons as in lines 81-83. Also, they should add something about the energetic brain and how astrocytes play a role in what’s known as “Tripartite synapse”. References:
https://pubmed.ncbi.nlm.nih.gov/10322493/ 
https://pubmed.ncbi.nlm.nih.gov/31318452/ 

Neurodegenerative diseases are no longer neuron only disorders since the role of glial cells have been extensively validated. For example, the water channel aquaporin 4 (AQP4), which is highly expressed on astrocyte endfeet, critically regulates water flux between blood and brain. AQP4 plays an essential role in regulating the recently discovered glymphatic system which is a waste clearance system that utilizes a unique system of perivascular channels, formed by astroglial cells, to promote efficient elimination of soluble proteins and metabolites from the central nervous system.

References:
https://www.ncbi.nlm.nih.gov/pmc/articles/PMC4252540/ 
https://pubmed.ncbi.nlm.nih.gov/30561329/ 
https://pubmed.ncbi.nlm.nih.gov/33004510/ 

Lines of evidence showed that CNS disorders, inflammation and vascular diseases caused by astrocytes, leading to brain edema and exacerbate neurodegenerative diseases. This has been recently been demonstrated by the breakthrough study by Kitchen et al Cell 2020 where they showed that targeting astrocytes following ischemia and hypoxia not only reduces edema but also stabilises the BBB/BSCB barriers and led to accelerated functional recovery compared with untreated animals. This role has been recently been confirmed by the work of Sylvain et al BBA 2021 which has demonstrated that targeting astrocytes is a viable therapeutic target using a photothrombotic stroke model. They have also shown a link to brain energy metabolism as indicated by the increase of glycogen levels.

References:
https://pubmed.ncbi.nlm.nih.gov/32413299/ 
https://pubmed.ncbi.nlm.nih.gov/33561476/ 

Moreover, authors need to mention the work by Qian et al. Nature 2020 where they have beautifully shown that the conversion of midbrain astrocytes to dopaminergic neurons, which provide axons to reconstruct the nigrostriatal circuit.

References:
https://pubmed.ncbi.nlm.nih.gov/32581380/ 

We agree with the reviewer's comments regarding the lack of depth in the role of cofilin in glial cells present in our review. We focused on neurons, but we agree with the reviewer that not only neurons that can trigger neurodegenerative processes. This has led us to modify the structure of the manuscript including two new points following the suggestions of the reviewer. In both points, a review is made of the role of cofilin in two very important processes, neuroinflammation (point 3.2) and myelination (point 3.3). The role of glial cells in both processes is clear and we have underscored its role. Glial alteration promotes many of the known neurodegenerative diseases.

We have also included in point 1 (line 78-90) a short introduction to glial cells and the important functions they perform in relation to the dynamic processes of the cytoskeleton. We thank the reviewer for all the references he/she has provided. We have tried to include many of them in the new version of the manuscript.

-End of discussion and towards the conclusion: NDs are yet incurable diseases. Author needs to point out to the recent advances in applying the use of high-throughput screening and computer-aided drug design as have been nicely reviewed by Aldewachi et al 2021 as they can provide a novel insight that can support target validation in future studies.

References:
https://pubmed.ncbi.nlm.nih.gov/33672148/ 

Following the recommendation of the reviewer, we have included a new paragraph in order to mention recent advances in applying the use of high-throughput screening and computer-aided drug design.

Reviewer 2 Report

The Review by Lapeña-Luzón et al discusses the role of cofilin in neurodegenerative disorders. 

Cofilin is an important protein in many cellular processes and very less is known about it in neurodegenerative disorders. This review provides readers new insights and some background for further research. 

The review is well written and up to date. The only thing which I find missing is authors failed to discuss activity of cofilin which is the most critical part. Phosphorylation at serine 3 residue of cofilin is very critical to govern its activity. Cofilin promotes the regeneration of actin filaments by severing preexisting filaments. The severing activity of cofilin is inhibited by LIMK or TESK phosphorylation at Ser3 of cofilin.

Authors should also discuss this to make it a complete review for readers.

Author Response

We are very grateful to the reviewer for their time and positive suggestions. The reviewer’s comments are shown in black and our replies are shown in red.

Reviewer 2:

The Review by Lapeña-Luzón et al discusses the role of cofilin in neurodegenerative disorders. 

Cofilin is an important protein in many cellular processes and very less is known about it in neurodegenerative disorders. This review provides readers new insights and some background for further research. 

The review is well written and up to date. The only thing which I find missing is authors failed to discuss activity of cofilin which is the most critical part. Phosphorylation at serine 3 residue of cofilin is very critical to govern its activity. Cofilin promotes the regeneration of actin filaments by severing preexisting filaments. The severing activity of cofilin is inhibited by LIMK or TESK phosphorylation at Ser3 of cofilin. Authors should also discuss this to make it a complete review for readers.

Following the reviewer's recommendations, we have expanded the section on cofilin regulation, showing the complex system of regulation of both the cofilin phosphorylation / dephosphorylation process and the proteins that carry it out (point 2, line 96-112).

Round 2

Reviewer 1 Report

Dear Editor,

The authors have successfully addressed the majority of my comments and concerns in order to improve the quality of the manuscript. 

I do believe that the corrections, new sections and updated references, have contributed to enhancing the clarity of the manuscript, which I can now endorse for publication following one minor edit.

All the best.

Minor:

-Authors forgot to mention and discuss the work of Kitchen et al. Cell 2020 and Sylvain et al 2021. This can be added to line 95 following the mention of role of glial cells or at line 318. I hereby add the comment again:

Lines of evidence showed that CNS disorders, inflammation and vascular diseases caused by astrocytes, leading to brain edema and exacerbate neurodegenerative diseases. This has been recently been demonstrated by the breakthrough study by Kitchen et al Cell 2020 where they showed that targeting astrocytes following ischemia and hypoxia not only reduces edema but also stabilises the BBB/BSCB barriers and led to accelerated functional recovery compared with untreated animals. This role has been recently been confirmed by the work of Sylvain et al BBA 2021 which has demonstrated that targeting astrocytes is a viable therapeutic target using a photothrombotic stroke model. They have also shown a link to brain energy metabolism as indicated by the increase of glycogen levels.

References:
https://pubmed.ncbi.nlm.nih.gov/32413299/
https://pubmed.ncbi.nlm.nih.gov/33561476/

Author Response

Once again we are grateful to the reviewer for his comments and we are pleased that he found the corrections appropriate. We also think that the manuscript has improved the quality thanks to the new sections that have been incorporated.

Regarding the references mentioned by the reviewer, we have decided that it is not necessary to include them, since they are not part of the main topic that is reviewed in the manuscript.